# Monitoring Human Milk β-Casein Phosphorylation and *O*-Glycosylation Over Lactation Reveals Distinct Differences between the Proteome and Endogenous Peptidome

**DOI:** 10.3390/ijms22158140

**Published:** 2021-07-29

**Authors:** Kelly A. Dingess, Inge Gazi, Henk W. P. van den Toorn, Marko Mank, Bernd Stahl, Karli R. Reiding, Albert J. R. Heck

**Affiliations:** 1Biomolecular Mass Spectrometry and Proteomics, Bijvoet Center for Biomolecular Research, Utrecht Institute for Pharmaceutical Sciences, University of Utrecht, 3584 CH Utrecht, The Netherlands; k.a.dingess@uu.nl (K.A.D.); i.gazi@uu.nl (I.G.); H.W.P.vandenToorn@uu.nl (H.W.P.v.d.T.); 2Netherlands Proteomics Center, 3584 CH Utrecht, The Netherlands; 3Danone Nutricia Research, 3584 CT Utrecht, The Netherlands; Marko.MANK@danone.com (M.M.); bernd.stahl@danone.com (B.S.); 4Chemical Biology & Drug Discovery, Utrecht Institute for Pharmaceutical Sciences, University of Utrecht, 3584 CG Utrecht, The Netherlands

**Keywords:** human milk, mass spectrometry, *O*-glycosylation, peptidomics, antimicrobial peptides

## Abstract

Human milk is a vital biofluid containing a myriad of molecular components to ensure an infant’s best start at a healthy life. One key component of human milk is β-casein, a protein which is not only a structural constituent of casein micelles but also a source of bioactive, often antimicrobial, peptides contributing to milk’s endogenous peptidome. Importantly, post-translational modifications (PTMs) like phosphorylation and glycosylation typically affect the function of proteins and peptides; however, here our understanding of β-casein is critically limited. To uncover the scope of proteoforms and endogenous peptidoforms we utilized mass spectrometry (LC-MS/MS) to achieve in-depth longitudinal profiling of β-casein from human milk, studying two donors across 16 weeks of lactation. We not only observed changes in β-casein’s known protein and endogenous peptide phosphorylation, but also in previously unexplored *O*-glycosylation. This newly discovered PTM of β-casein may be important as it resides on known β-casein-derived antimicrobial peptide sequences.

## 1. Introduction

Human milk is a diverse biofluid, providing unique nutritional, non-nutritional, and bioactive components to the infant. There are many factors that affect the composition of human milk throughout lactation, including maternal factors such as diet, age, and body mass index and infant factors such as gestational age and sex, as well as a range of influences from the environment, physiology, and behavior [1,2,3]. Among the many different components (sugars, lipids, etc.) of human milk, proteins and peptides are key important nutritional and bioactive molecular factors. The human milk proteome is continuously changing in composition throughout lactation, from colostrum to mature milk. For example, the overall protein content declines from 20 g/L down to 10 g/L and the whey/casein ratio shifts from 90/10 to 60/40 [2,4]. To gain deeper insights into the composition of human milk, personalized profiling offers unique opportunities, especially regarding the human milk proteome, encompassing both proteins and endogenous peptides. As human milk composition is donor-specific, personalized profiling strategies bring us closer to methodologies assisting individualized health care, taking into account a person’s unique characteristics [5]. For human milk, this can lead to novel insights into the maternal–breastmilk–infant triad [6].

The human milk proteome is comprised of not only proteins, but also endogenous peptides derived from proteins upon proteolysis within the mammary gland. Many of these peptides are bioactive in their own regard and carry functions independent of their proteins of origin. These activities include antimicrobial, angiotensin converting enzyme (ACE) inhibition, dipeptidyl peptidase IV (DPP-IV) inhibition, opioid agonist and antagonist activities, immunomodulation, mineral binding, and antioxidative functions [7]. It is estimated that the endogenous peptidome contributes up to 2–3% of the total protein concentration [8,9,10]. The majority of human milk endogenous peptides are derived from abundant milk proteins—such as caseins, osteopontin, and polymeric immunoglobulin receptor [11,12,13,14,15]. The main protein contributing to the peptidome is β-casein, making up approximately 50% of all endogenous peptides [11,12,13,14,15]. Therefore, it is critical to know the β-casein derived peptidome in detail, including how this may change dynamically over lactation, and how it is possibly modified by post-translational modifications (PTMs) such as phosphorylation and glycosylation, which may affect functionality. For the latter, it is also intriguing to know if such PTM features are distinctive between the proteome and peptidome.

β-casein is a 226 amino acid (including the signal peptide), 25 kDa highly-phosphorylated protein. It makes up the majority of the casein micelle in human milk, followed by κ-casein and small amounts of α_s1_-casein. Structurally, a casein micelle consists of α_s1_-casein and β-casein forming an inner core, with κ-casein making up the outer glycosylated layer that stabilizes the micelle, see Figure 1 [16,17]. The inner region of the casein micelle remains uncertain, resulting in an undetermined structure of β-casein. Even so, the lack of disulphide bridges suggests the protein is likely to not have a tightly defined secondary or tertiary structure, whereas the abundant prolines in the sequence predispose it to an open conformation [16]. This overall open structure makes the protein readily accessible to proteolytic cleavage. Plasminogen, its activators, and its inhibitors, are reported to be associated with the casein micelle and are predominantly responsible for β-casein degradation in the mammary gland, and thus for producing the putative bioactive peptides [18].

The main functions of casein proteins in human milk are to provide essential amino acids [19] and to bind and transport divalent cations, such as calcium and zinc, facilitating the absorption of these nutrients in the gut of the nursing infant [20]. The overall abundant phosphorylation of β-casein is thought to contribute to these functions. Moreover, there has also been antimicrobial functionality attributed to the C-terminal peptide derived from β-casein, (_200_QELLLNPTHQYPVTQPLAPVHNPISV_226_), in recent human milk peptidomics studies [14,21,22,23,24,25,26]. Additionally, this peptide has been found to survive infant digestion, indicating that it could exert antimicrobial functionality in the distal part of the infant gut [24]. Interestingly, all these studies use one source for justification of antimicrobial functionality of this peptide [27]. Namely, Minervini et al., tested the functionality of milk protein hydrolysates from six mammalian species and found that one particular hydrolyzed human milk sodium caseinate fraction containing the aforementioned peptide (and potential co-isolated variants thereof) showed considerable antimicrobial functionality against *Escherichia coli* (*E. coli) F19* [27]. Very recently, the Minervini et al. results have been replicated by one group, which has published several studies confirming the antimicrobial functionality of various peptides across the C-terminus of β-casein [28,29,30].

Post-translational modifications (PTMs) are important regulators of proteins, often determining a protein’s functionality. Phosphorylation is a widespread and critical PTM, and as such it is not surprising that β-casein exhibits differing functionality based on its varying degrees of abundant phosphorylation. Studies have shown that casein phosphorylation is critical for the formation of casein micelles and the subsequent delivery and absorption of calcium, phosphate and other minerals in the infant gut [31,32]. Additionally, protein glycosylation, one of the most prominent and complex PTMs of the human milk proteome, is also important when considering functionality. Glycosylation refers to the covalent attachment of oligosaccharides on the protein backbone as *N-* or *O*-glycosylation. The most well-studied of these is *N-*glycosylation, in which an oligosaccharide chain is attached to the protein via the amide group of an Asn residue within a defined sequon (Asn-Xxx-Thr/Ser, Xxx not being Pro) [33]. With *O*-glycosylation, the oligosaccharide chain is attached to the protein backbone via the oxygen of a hydroxylated amino acid, mainly Ser or Thr [34]. Importantly, it is estimated that up to 70% of the identified major abundant human milk proteins are glycosylated [35]. Notably, no glycosylation features have been reported so far for β-casein.

The diversity and abundance of milk glycan species is important to consider, as these glycosylated compounds comprise free oligosaccharides, glycoproteins, glycopeptides, and glycolipids [36]. Moreover, these milk glycans are energetically costly for the mother to produce and predominantly indigestible by the infant, suggesting important alternative roles [37]. The delivery of these functional milk components changes throughout lactation to meet the developing needs of the infant, such as their innate immune system and gut microbiota. For instance, milk glycans work to shape the infant’s intestinal ecology, binding or inhibiting binding of bacteria, viruses, and toxins to intestinal epithelial cells, enriching the gut microbiota with bifidobacteria, reducing inflammation, and promoting intestinal epithelial barrier function [37].

Recent advancements in analytical mass spectrometry have made it possible to characterize a variety of PTMs and how they dynamically change over time. Understanding this dynamic change is essential for gaining insights into the functionality of these proteins. Currently, there is a lack of understanding how dynamic protein changes affect the functionality of both the proteome and endogenous peptidome in human milk. We sought to investigate this in a protein specific manner for multiple PTMs in a longitudinal personalized profiling approach. We now show, for both the β-casein proteome and endogenous peptidome, how phosphorylation and *O*-glycosylation change over the first 16 weeks of lactation in two individual donors. In doing so, we have uncovered for the first time two *O*-glycosylation sites on the C-terminus of β-casein, in the peptide that has previously been shown to be antimicrobial.

## 2. Results

An initial in-depth proteomics screen of our data revealed evidence that the human milk protein β-casein seemingly harbored *O*-glycosylation modifications, in addition to being highly phosphorylated. As *O*-glycosylation has yet to be described for this protein, we focused on characterizing β-casein as a novel *O*-glycoprotein with distinct glycan modifications in the intact protein and in the endogenous *O*-glycopeptidome. Longitudinal personalized profiling was performed to detect human milk peptides carrying *O*-glycans, wherein milk from two individual donors was assessed for both the proteome and peptidome. The overall experimental workflow is depicted in Appendix A. In brief, milk samples were collected longitudinally from two mothers across colostrum (<72 h), transitional (>3–15 days), and mature (>16 days) lactational stages [38,39,40], i.e., at weeks 1, 2, 3, 4, 6, 8, 10, 12, and 16. Standard protein and endogenous peptide extraction methods were applied as described before [41], and samples were analyzed by label-free shotgun glycoproteomics and glycopeptidomics approaches.

Typically, to analyze glycopeptides, one first enriches them from the large pool of non-modified peptides. However, here we show that making use of advances in MS, namely hybrid fragmentation approaches such as HCD-pd-sHCD and HCD-pd-EThcD, large-scale characterization of thousands of intact glycopeptides is feasible [42,43,44], even from highly complex samples such as human milk. In this work, we focused our analysis on β-casein, with the aim to get a comprehensive view of all its proteoforms and endogenous peptide modifications. β-casein is an important and ideal protein candidate to investigate these PTMs at the proteome and peptidome level, as it makes up a large percentage of both. In terms of abundance, 65% of the endogenous peptidome originated from caseins (α_s1_-, β-, and κ-), with β-casein making up 50% of the total peptidome, while in the proteome the caseins only made up about 30% of total protein abundance [41].

### 2.1. Mass Analysis of Intact β-Casein Proteoforms

Initially, intact protein mass spectrometry was performed to evaluate the scope of all β-casein proteoforms in human milk. From this analysis we were able to annotate a total of 16 β-casein proteoforms. These included the six previously described phosphoproteoforms—i.e., β-casein 0–5P—and ten newly-detected glycoproteoforms (Supplemental Appendix A). Figure 2 depicts the deconvoluted mass spectra of the β-casein proteoforms detected in the whole milk of donor 2. While all β-casein phosphoproteoforms in the range of 0–5P could be detected, the 2P and 4P forms were most abundant. This is in line with previous studies that also found 2P and 4P to be the most abundant of the six known β-casein proteoforms [45,46]. Phosphosite occupancy could potentially affect the ability β-casein to bind calcium. We observed a decrease of phosphosite occupancy over lactation for both donors, both in the proteome and in the endogenous peptidome. According to Neville et al. (1994), the total calcium content of human milk decreased during lactation, with the absolute concentration of the soluble calcium remaining stable [47]. This strongly indicates that particularly the protein-bound calcium, rather than total calcium, decreased. This is line with a decrease in the capacity of β-casein to bind calcium, possibly due to the decrease in the degree of its phosphorylation across lactation. We observed an overall decline in phosphorylation across lactation as well, Figure 2.

The glycoproteoforms were detected as minor constituents and were identified by mass differences of the non-glycosylated proteoforms and corresponding glycan residues. Glycoproteoforms of the 0 and 1P forms of β-casein can be observed and the mass differences in this case could be matched to the glycan masses of N_1_ and N_1_H_1_ in the minor proteoforms depicted across lactation in Figure 2 (F = deoxyhexose, H = hexose, N = *N*-acetylhexosamine, S = *N*-acetylneuraminic acid). The presence of minor glycoproteoforms was confirmed in both donors across lactation (Figure 2, data only depicted for donor two), revealing unambiguously that β-casein is not only a phosphoprotein, but also a glycoprotein. For further confirmation of β-casein glycosylation and for PTM site localization the samples were next analyzed by bottom-up LC-MS/MS.

### 2.2. Data Analysis Strategy for Bottom-Up Mass Spectrometry

Using a bottom-up mass spectrometry approach, we identified a total of 44,667 non-modified and 2286 glycosylated β-casein peptide-to-spectrum matches (PSMs, a measure of correctly annotated spectra) from the proteome data, and 48,295 non-modified and 2241 glycosylated from the peptidome data. These metrics are from accumulated data for both donors and all time points after data curation presented in Appendix A. After strict filtering criteria, an average of 5 percent of all PSMs, coming from both donors across all time points, could be attributed to glycosylation in the proteome and peptidome, 2286 and 2241 PSMs, respectively. Curation criteria were selected to reduce potentially false PSMs, determined by the count of reverse sequences identified by Byonic.

The first step in the curation criteria was to remove PSMs from the signal peptide, amino acids 1–15, leaving only PSMs generated from the mature protein. Second, only PSMs matched to spectra of non-negligible error probabilities were accepted, i.e., |log Prob| ≥ 1.5, Byonic score ≥ 150 and Delta Mod score > 5. Next, we used the number of PSMs as a proxy for the abundance of the given PTMs in the proteome and peptidome data. One can argue that highly abundant PTMs, compared to those with lower abundance, will have increased chromatography elution time widths, more charge states that can be detected, more structural isomers that can be measured separately, a higher chance of triggering MS2, an improved chance of identification by a search engine, and importantly, it will be detected on more low-abundance peptide variants. While the absolute number of detected PSMs cannot be directly compared between the proteome and peptidome datasets, within each dataset the observed trends and relative occupancy ratios are expected to be representative.

After applying strict automatic filtering criteria, we additionally inspected each potential phospho- and glycosite for the presence of the correct peptide fragments and fragments corresponding to phosphorylation and/or glycosylation. All automatically annotated glycosites on the N-terminus of the protein were found to lack supportive fragmentation evidence and were excluded from further analysis. We show in Appendix A (donor one, week 16, EThcD) the negative characteristics of spectra that were not included in the final interpretation of glycopeptide PSMs. Negative characteristics included the following: abundant peaks without annotation, nonsensical or high numbers of glycan species that are not supported by fragment series (e.g., more than triply glycosylated, but without corresponding oxonium ions) and an absence of peptide fragments with retained modifications (either for glycosylation or phosphorylation).

### 2.3. β-Casein Phosphorylation Analysis

The total PTM PSMs for both phosphorylation and glycosylation were found to be variable between the two donors and across lactation (Appendix A). We identified confidently five phosphosites (Ser/Thr) spanning the N-terminal peptide _16_RE**T**IE**S**L**SSS**EESIT_30_, with the detected phosphorylation sites in bold (Figure 3, Appendix A). In accordance with previous literature [31,45,46,48,49,50], we found this N-terminal phosphorylation cluster to have up to all five of the aforementioned phosphosites occupied with varying degrees of stoichiometry between donors, across lactation, and relative to the proteome or peptidome (Figure 4). Overall, site Ser24 showed the highest number of PSMs (e.g., the mean phosphosite PSMs across all time points being 197 and 663 for respectively the proteome and peptidome for donor one), followed by Ser25, Ser23, Ser21, and Thr18 regardless of the donor, lactational stage, proteome or peptidome data (Figure 4, Appendix A). This overall observed trend in phosphosite occupancy is in line with literature data [48,51]. The identified phosphosites follow the Ser/Thr-Xxx-Glu/pSer sequence motif recognized by the FAM20C kinase, which has been identified as the Golgi casein kinase responsible for this phosphorylation [52,53]. Accordingly, Ser24 and Ser25, also fitting this motif, were found to be the most highly phosphorylated (~70–85% per site). These two phosphosites also displayed differing degrees of phosphorylation across lactation and between the proteome and the endogenous peptidome. For instance, in the proteome and the peptidome, sites Ser24 and Ser25 were observed to have site occupancies of >90% at early lactation, with a gradual decline to 65% at later stages of lactation (Figure 4). However, the number of occupied phosphosites differed between the peptidome and the proteome, with respectively 4–5 and 2–3 phosphosites occupied at any given time, indicating that β-casein endogenous peptides were generally more phosphorylated than the β-casein protein (Figure 4). Additionally, there was a high degree of individual variability in the degree of phosphosite occupancy between donors across lactation. For instance, in the peptidome, donor one had the highest phosphosite occupancy on sites Ser24 (mean 85%) and Ser25 (mean 84%) at any given time, compared to donor two Ser24 (mean 72%) and Ser25 (mean 70%). However, the dynamic change in site occupancy was less dramatic over lactation in donor one than donor two, only changing by 17% in donor one versus up to 34% in donor two, Appendix A. Similar trends were also observed in the proteome indicating that phosphosite occupancies can be donor-dependent.

While the N-terminal stretch of β-casein is heavily decorated by several phosphorylations, we found no convincing evidence of phosphorylation occurring on any other Ser or Thr residues in the protein sequence in both the proteome and the peptidome data (Figure 4, Appendix A). This is in accordance with the fact that no other Ser or Thr residues in the amino acid sequence of the protein are found to fit the FAM20C phosphorylation motif.

### 2.4. β-Casein Glycosylation Analysis

We found convincing evidence that the C-terminal stretch of β-casein contained *O*-glycans, both in the proteome and peptidome data. As this finding is novel, and *O*-glycosylation quite difficult to annotate, we recorded *O*-glycopeptide fragmentation by EThcD, as this provides often more confident assignments [54]. An illustrative set of annotated EThcD spectra, spanning the C-terminal ladder peptide (_197_LLNQELLLNPTHQYPVTQPLAPVHNPISV_226_), are depicted in Figure 5, illustrating fragment ions indicative of the peptide backbone, carrying small neutral, sialylated, and fucosylated *O*-glycan species. Predominantly, in both the proteome and peptidome data, the sites Thr207 and Thr214 were found to be occupied by a variety of several different *O*-glycan species (Figure 6, Appendix A). For donor one specifically, across all time points for Thr207 and Thr214 we identified a total of 813 (mean 90, 12% occupancy) and 1082 (mean 120, 17% occupancy) *O*-glycopeptide PSMs in the proteome and 619 (mean 69, 12% occupancy) and 852 (mean 95, 17% occupancy) *O*-glycopeptide PSMs in the peptidome (Figure 4, Appendix A). For donor two, across all time points for Thr207 and Thr214 we identify a total of 465 (mean 52, 10% occupancy) and 709 (mean 79, 16% occupancy) *O*-glycopeptide PSMs in the proteome and 546 (mean 61, 13% occupancy) and 723 (mean 80, 17% occupancy) *O*-glycopeptide PSMs in the peptidome (Figure 4, Appendix A, Appendix A).

Notably, the stoichiometry of glycosylation changed throughout lactation, a feature even more intense in the peptidome than the proteome. For instance, for both donors in the proteome, glycosite occupancy remained rather consistent, with Thr207 at 10% and Thr214 at 16% (Appendix A). However, for both donors in the peptidome, there was a clear declining trend in these site occupancies over lactation. An exception to this trend was observed in donor one at week 6 which had site occupancies spike back to the values observed at week 1. This aberrant milk composition at one point in time from donor one corresponded to a potential maternal infection, as previously reported in detail [41]. Interestingly, the glycans occupying these two sites are predominantly made up of three species—N_1_H_1_, N_1_, and N_1_F_1_ (in order of highest to lowest PSM counts)—regardless of being identified in the proteome or peptidome Figure 6.

Although quantification using PSMs showed distinct trends between donors and across time, we aimed to further validate our *O*-glycan quantification by comparing precursor ion MS1 areas using Skyline [55,56]. The MS1 integration indicated that, in both the proteome and the peptidome data, glycopeptide masses could be detected that were comparative with Byonic’s assignment of one and two occupied glycosylation sites (Figure 7, Appendix A). Furthermore, the MS1 integration verified an overall approximate 5% glycosylation occupancy, in line with the fragmentation data derived by Byonic. It also supported that the most abundant *O*-glycosylation occurred at early lactation and gradually declined over time. This change could be seen for multiple ladder peptides ranging from _190_AVPVQALLLNQELLLNPTHQIYPVTQPLAPVHNPISV_226_ to _200_QELLLNPTHQIYPVTQPLAPVHNPISV_226_ (Figure 7, Appendix A). Overall, MS1 area integration yielded highly comparable trends with PSM-based quantification (Appendix A), with early lactation resulting in more overall glycosylation than later lactation. Indicating that both analysis strategies were viable to use for judging *O*-glycan occupancy on β-casein.

We could also trace back non-modified ladder peptides spanning this sequence, from _200_QELLLNPTHQYPVTQPLAPVHNPISV_226_ down to _218_APVHNPISV_226_ (Appendix A). All of these peptides spanning the same sequence exhibited different concentrations and distinctive trends in abundance between donors and across lactation. Although currently not known, all these peptides may have variable biological activity. Like the bottom-up site occupancy data, again donor one week 6 becomes an interesting time point, where the MS1 data for the reported antimicrobial peptide does not follow a clear lactational trend. Rather, at this aberrant time point, this supposed antimicrobial peptide, and many of the other ladder peptides, on average goes substantially up relative to all other time points (Appendix A).

## 3. Discussion

The overall analysis of PTMs in complex biofluids by mass spectrometry is challenging due to the high dynamic range of proteins, overall low abundance of peptides with PTMs relative to non-modified peptides, and potential ion suppression of modified peptides. To overcome these challenges sample preparation methods have focused on enriching for a particular PTM of interest, such as phosphorylation or *N-*glycosylation. This in turn limits the analysis to these chosen PTMs, not allowing a full proteome-wide approach, but rather requiring the analysis of the proteome and the enriched PTM fraction of the proteome in parallel. Even with this approach, one is usually limited to the most abundant proteins in a given sample. There have been steps to implement these advances in the analysis of human milk PTMs such as phosphorylation [48] and *N-*glycosylation [57,58] over the course of lactation, showing the dynamic variability of these PTMs over time. Even with these advancements, additional PTMs such as *O*-glycosylation have thus far been largely neglected. However, recent advancements in mass spectrometry now enable the analysis of these PTMs simultaneously without prior sample enrichment. From two donors over the first 16 weeks of lactation, we observed gradual temporal changes in the total PSM counts of phosphorylated and *O*-glycosylated peptides on both the protein β-casein and its endogenous peptidome. Interestingly, the endogenous peptidome consisted overall of more phospho- and *O*-glycopeptides than the proteome, indicating that PTMs may contribute to a rich source of functional endogenous peptides that has received limited attention.

### 3.1. Phosphorylation Differences between β-Casein and Its Peptidome

The degree of phosphorylation on β-casein not only regulates the size of the casein micelle, it also influences the function of the protein [31,51,59]. This, in turn, affects the overall digestibility, release, and absorption of functional proteins and peptides in the infant intestinal tract. The phosphosites of human milk caseins have been extensively studied [31,48,49,50] and even the site occupancy over the first month of lactation has been well-characterized in terms of the intact protein [48], but work on the peptidome has lagged behind. Previous studies have shown that the preferred phosphorylation order is Ser24 > Ser25 > Ser23 > Ser21 > Thr18, with a preference for four sites being occupied at any given time throughout lactation [48,51]. Our data are in line with these reports regarding the preferential order of phosphosites, but our data suggests that occupancy of more than three phosphosites predominantly occurs on the endogenous peptides and not the intact protein. Even in the proteome data we found that the third phosphosite was only occupied about 29–33% of the time, whereas in the peptidome it was likely that four phosphosites were occupied at least 33–53% of the time.

The degree of phosphorylation of β-casein is known to directly affect its function. For instance, different phosphoproteoforms are better at inhibiting *H. influenzae* than others [51]. This study found that the most common phosphoproteoforms of β-casein, by concentration, were tetra- > di- > non- > mono- > tri- > penta-phosphorylated. Furthermore, they found that in an anti-adhesion assay, the tri-, tetra-, and penta-phosphorylated forms of β-casein exhibited more than 60% inhibition of *H. influenzae* [51]. Other effects of phosphorylation functionality have been less well-studied; for instance, how the differing degrees of phosphorylation of β-casein affect casein peptide release both in the mammary gland and in the infant’s stomach. Currently, there is little evidence on endogenous phosphopeptidoforms and their bioactive functions, and the data provided here for β-casein may prove useful for future functional characterization.

### 3.2. O-Glycosylation Changes Across Lactation in the β-Casein Protein and Its Peptidome

We report here, for the first time, the presence of *O*-glycans on β-casein, in particular on the C-terminal stretch at sites Thr207 and Thr214. We observed individual specific and longitudinal dynamic changes in the site occupancy of *O*-glycosylation across the proteome as well as the peptidome data. Even with these dynamic differences, consistently both Thr207 and Thr214 contained *O*-glycans. The most dominant detected glycan moieties were N_1_H_1_. However, the second most abundant glycan species seems to be different between the intact protein and endogenous peptides, irrespective of donor or lactational stage, in which the proteome data contained more N_1_ glycans, and the peptidome data had substantially more N_1_F_1_ glycans. The individual specific nature of human milk glycosylation has been previously observed in *N*-glycoproteomics studies as well [57,58].

Our finding of *O*-glycans on the C-terminal stretch of β-casein is important in the context of the work by Minervini et al., which determined that the C-terminal β-casein derived peptide (_200_QELLLNPTHQIYPVTQPLAPVHNPISV_226_) has antimicrobial functionality [27]. Minervini et al. generated this peptide by preparing sodium caseinate from human milk and subsequently hydrolyzing it with the *Lactobacillus helveticus* PR4 proteinase. The purified peptide fraction was found to exhibit biological antimicrobial activity. It is highly probable that the glycosylated forms of the peptide were unknowingly co-enriched and contributed to the observed bioactivity. While the work of Minervini et al. focused on the unique peptide generated by the specific *Lactobacillus helveticus* PR4 proteinase, another group has recently published on several ladder series peptides related to the initially reported antimicrobial peptide, _200_QELLLNPTHQIYPVTQPLAPVHNPISV_226_, for which activities were tested in cell culture assays against varying bacterial species with synthetic (non-modified) peptides. Specifically, peptides derived from the C-terminal part of β-casein were found to have antimicrobial functionality against specific bacterial species: peptides 201–220 and 213–226 had activity against *S. aureus* and *Y. enterocolitica* [28,29]; peptide 211–225 had activity against *E. coli* and *Y. enterocolitica* [30]. Interestingly, in all of these studies, the varying ladder peptides showed the same mode of antimicrobial activity, by membrane disrupting mechanisms, but not by binding to intracellular nucleic acids. While these studies used concentration ranges consistent with our own data, 0.5–20 µg/mL, we only observe higher concentrations during the first two weeks of lactation and again in donor one at week 6, which relates to a suspected period of infection [41]. Furthermore, these studies indicate that not only the specific peptide reported by Minervini et al., but in fact a series of C-terminal ladder peptides as also detected in our study are all biologically active. Therefore, it is reasonable to assume that adding *O*-glycosylation to the C-terminal ladder series of peptides from β-casein may influence their bactericidal activity. Therefore, while synthesis of *O*-glycosylated peptide standards is still a major challenge such antimicrobial assays should ideally take into account the here-reported *O*-glycosylation features.

## 4. Materials and Methods

### 4.1. Human Subjects and Milk Samples

Details of subjects, bottom-up proteomics, and mass spectrometry (MS) methods have been described in detail before [41]. Longitudinal human milk samples were collected from two healthy donors at weeks 1, 2, 3, 4, 6, 8, 10, 12, and 16 postpartum. Samples were collected according to standardized human milk handling conditions [60]. All samples were collected into 2 mL Eppendorf tubes containing protease and phosphatase inhibitors as 1/9 of the collection volume, Complete Mini EDTA-free (Roche) and PhosSTOP (Roche), respectively. Samples were transferred back to the lab on dry ice and stored at −80 °C until analysis. Written informed consent was obtained from both donors prior to sample collection. All samples used were donated to Danone Nutricia Research in accordance with the Helsinki Declaration II.

### 4.2. Whole Milk Proteolytic Digestion

The in-solution digestion of whole milk was adapted from previous methods [61] and detailed in full [41]. Briefly, for bottom-up proteomics, whole milk proteins were denatured, reduced, and alkylated, followed by digestion with trypsin (Sigma-Aldrich, Steinheim, Germany) at 37 °C for 16 h. The resulting peptides were purified by solid phase extraction (SPE) using Oasis PRiME HLB 96-well plates (Waters, Etten-Leur, The Netherlands), according to the instructions of the manufacturer, dried by SpeedVac and stored at −80°C until analysis. Prior to LC-MS/MS analysis the dried peptides were reconstituted in 2% formic acid (FA) to achieve an injection of 800 ng material on column.

### 4.3. Skimmed Milk Isolation of Endogenous Peptides

Methods for skimmed milk peptide extraction have been previously described in detail [62]. Briefly, samples were defatted by centrifugation, 15 min at 1500× *g* at 4 °C, followed by removal of proteins and impurities by precipitation with 20% trichloroacetic acid (TCA) (Sigma-Aldrich, Steinheim, Germany). TCA was added as equal volumes of milk and 20% (*v*/*v*) TCA, resulting in a final TCA concentration of 10% (*v*/*v*). The peptides extracted in the supernatant were purified by SPE using Oasis PRiME HLB 96 well plates (Waters, Etten-Leur, The Netherlands), dried by SpeedVac and stored at −80°C until analysis. Prior to LC-MS/MS analysis the samples were reconstituted in 0.1% TFA and then further diluted in 2% FA to achieve an injection of 800 ng of material on column.

### 4.4. High-Pressure Liquid Chromatography Tandem Mass Spectrometry Glycopeptide Analysis

Tryptic and endogenous peptides were injected on column as 800 ng, analyzed using an Agilent 1290 Infinity HPLC system (Agilent Technologies, Waldbronn, Germany) coupled on-line to an Orbitrap Fusion mass spectrometer (Thermo Fisher Scientific, San Jose, CA, USA) using a 120 min and 60 min gradient for tryptic and for endogenous peptides respectively. All samples were ran as MS triplicates with varying MS/MS fragmentation types as higher-energy collisional dissociation (HCD), HCD-product-dependent stepping collision energy HCD (HCD-pd-sHCD; see Appendix A for triggering ions) and HCD-product-dependent electron-transfer/higher-energy collision dissociation (HCD-pd-EThcD).

The 60 min gradient was run as follows: 100% solvent A for 5 min, 13–44% solvent B for 40 min, 44–100% solvent B for 3 min, 100% solvent B for 2 min, and 100% solvent A for 10 min. The 120 min gradient was run with the same percentages of solvents A and B over double the gradient time. Peptides were ionized using a 2.0 kV spray voltage. For the MS scan, the mass range was set to *m/z* 350–2000 for a maximum injection time of 50 ms at a mass resolution of 60,000 and an automatic gain control (AGC) target value of 4 × 10^5^ in the Orbitrap mass analyzer. The dynamic exclusion time was set to 30 s for an exclusion window of 10 ppm with a cycle time of 3 s. Charge-state screening was enabled, and precursors with 2+ to 8+ charge states and intensities > 1 × 10^5^ were selected for tandem mass spectrometry (MS/MS). HCD MS/MS (*m/z* 120–4000) acquisition was performed in the HCD cell, with the readout in the Orbitrap mass analyzer at a resolution of 30,000 (isolation window of 1.6 Th) and an AGC target value of 5 × 10^4^ or a maximum injection time of 50 ms with a normalized collision energy (NCE) of 30%. If at least three oxonium ions of glycopeptides (Appendix A) were observed, HCD-pd-sHCD or HCD-pd-EThcD MS/MS on the same precursor was triggered (mass tolerance of 20 ppm) and fragment ions (*m/z* 120–4000) were analyzed in the Orbitrap mass analyzer at a resolution of 30,000, AGC target value as 400% the standard value, or a maximum injection time of 250 ms. Product-dependent sHCD was performed at NCEs of 10%, 25% and 40%. Product-dependent EThcD was performed at supplemental collision energy of 25%.

### 4.5. Glycopeptide Identification

All the raw files obtained for the glycopeptide identification were processed in Byonic (Protein Metrics Inc, v. 3.9.4, Cupertino, CA, USA) searching against a human β-casein protein database (UniProtKB accession P05814, downloaded 22 September 2020) with the following search parameters: non-specific digestion with, precursor ion mass tolerance, 10 ppm; fragmentation type, both HCD & EThcD; fragment mass tolerance, 20 ppm; no fixed modifications were included for the proteome or the peptidome (as the mature protein does not contain cysteins, carbamidomethylation is not necessary to consider); variable modifications: methionine oxidation, phosphorylation of serine and threonine as 7 common. For glycan analysis, we used a Byonic database of nine core 1 glycans (Appendix A). The list of nine glycans was defined following broader pre-searches using the built-in Byonic database of 70 human *O*-glycans. From the curated list of glycans, we acknowledge two limiting factors of our analysis approach; firstly, as we collected compositional information we principally do not distinguish between isomeric monosaccharides, e.g., GalNAc and GlcNAc residues; secondly, peptides which have been assigned with multiple *O*-glycans could instead carry one large glycan or a mixture of these mass-matching possibilities. The maximum number of precursors per scan was set to one and the FDR as 1%. Data was further curated with non-negligible error probabilities |log Prob| ≥ 1.5, Byonic score ≥ 150, and Byonic Delta Modification score > 5 deemed acceptable. Additionally, the signal peptide was removed, so only peptide sequences starting at amino acid position 16 (of the full protein sequence) were considered. Remaining reverse hits (<1%) were removed for subsequent data analysis. Determination of curation criteria is detailed in Appendix A.

Data processing was done with Anaconda3 (2020.07) distribution of Python 3.8.3. Following data processing, all figures were generated with R version 3.4.2, using ggplot2 (version 2.2.1).

### 4.6. Intact Protein Analysis by Mass Spectrometry

#### 4.6.1. Preparation of Human Milk Proteins for Intact Protein LC-MS and LC-MS/MS

Samples were prepared for a final protein concentration of 4 µg/µL. Skimmed milk samples, 40 µL, were prepared by adding 20 µL Milli-Q water followed by the addition of 30 uL of 100 mM tris(2-carboxyethyl)phosphine hydrochloride (TCEP) and finally 10 µL of 10% FA. Vortexing was performed in between each reagent addition. Samples were then incubated for 1 h at 60 °C under constant mixing. Following incubation, samples were filtered through a 0.45 µm syringe filter (Waters, Etten-Leur, The Netherlands) and stored at −20 °C until analysis. Prior to analysis the samples were diluted 8 times with 2% FA and a volume of 2 µL containing 1 µg of protein was used for the LC-MS/MS.

#### 4.6.2. Intact Protein LC-MS and LC-MS/MS Analyses

Chromatographic separation of intact protein samples was conducted on a Thermo Scientific Vanquish Flex UHPLC (Thermo Fisher Scientific, Germering, Germany) system equipped with MAbPac RP 1 × 150 mm column (Thermo Fisher Scientific, Germering, Germany). An amount of 1 µg of protein was loaded on the column heated to 40 °C. LC-MS/MS runtime was set to 27 min with flow rate of 100 µL/min. Gradient elution was performed using mobile phases A (water/0.1% FA) and B (ACN/0.1% FA): 10% B for 5 min, 10–31% B for 1 min, 31–41% B for 14 min, 41–95% B for 1 min, 95% B for 1 min, 95–10% B for 1 min, and 4 min column equilibration back to 10% B. All intact protein MS experiments were performed on an Orbitrap Fusion Lumos Tribrid mass spectrometer (Thermo Fisher Scientific, San Jose, CA, USA) set to Intact Protein Mode with Low Pressure setting. For analysis of intact proteins, three methods were employed: low-resolution MS, high-resolution MS, and high-resolution MS/MS. The low- and high-resolution MS approaches were used for full MS acquisition and had resolutions (at *m/z* 200) of 7500 and 120,000, respectively. In the high-resolution MS/MS approach, the resolution parameter was defined at 120,000 for both full MS and data dependent MS/MS. Full MS scans in all methods were acquired for the range of *m/z* 400–3000 with AGC target set to 2.5 × 10^6^ (250%). Maximum of injection time was defined at 250 ms with 5 µscans averaged for each scan. Data-dependent strategy was set to three scans per cycle. Selected ions were isolated with 4 Th window. Ion activation was set to ETD with a reaction time of 32 ms, reagent target of 1e6 and maximum injection time of 200 ms. All the data dependent MS/MS scans were recorded within the mass range of *m/z* 300–4000 with AGC target set to 2 × 10^7^ (2000%) and maximum of injection time defined at 250 ms. One µscan was acquired.

#### 4.6.3. Database Generation for Intact Protein Analysis

Database searching for intact protein LC-MS/MS analysis of human milk proteoforms was performed using the human β-casein (accession P05814) entry XML file downloaded from UniProtKB on 22 September 2020. The database imported from the XML file into ProSightPC 4.1 was treated as follows: initiator methionine removal, N-terminal acetylation as well as other PTMs contained in the XML file were allowed; up to 13 features and maximum proteoform mass of 70 kDa were allowed per sequence.

#### 4.6.4. Proteoform Library Generation for Matching of Intact Masses from LC-MS

A library of proteoforms was generated to include human β-casein proteoforms with phosphorylation in the range of 0–5 phosphorylated amino acid residues per sequence. For each phosphoproteoform, glycoproteoforms with 0–2 *O*-glycans were generated. The *O*-glycans considered are described in Appendix A. Proteoform monoisotopic and average masses were calculated based on the amino acid sequence of the mature protein (UniProtKB accession P05814, downloaded 22 September 2020, fragment 16-226) and the additional masses of the respective PTMs.

#### 4.6.5. Data Analysis for Intact Mass LC-MS and LC-MS/MS

Isotopically resolved and unresolved spectra obtained in intact protein LC-MS experiments of human milk proteins were deconvoluted with the BioPharma Finder 3.2 Software using Xtract or ReSpect algorithms (Thermo Fisher Scientific, San Jose, CA, USA), respectively. The Xtract parameters were as follows: signal-to-noise (S/N) threshold 3; *m/z* range 400–3000, charge range 2–50, and minimum number of detected charge states of 3. The source spectra were generated using the sliding windows algorithm with the following parameters: merge tolerance of 30 ppm, maximum retention time gap of 1 min and a minimum number of detected intervals of 3. ReSpect parameters: precursor *m/z* range 400-3000 and deconvolution mass tolerance 20 ppm. The source spectra were generated using the same sliding windows algorithm detailed for Xtract.

Automatic searches were performed using the Thermo Proteome Discoverer software (version 2.4.0.305, San Jose, CA, USA) with use of ProSightPD nodes for the high-resolution MS/MS experimental workflow. Two ProSightPD Annotated Proteoform nodes and the ProSightPD Subsequence Search node were run in parallel at 20, 500, and 20 ppm precursor mass tolerances, respectively. Fragment mass tolerance was in all three cases set to 20 ppm.

The deconvoluted monoisotopic masses from the high-resolution LC-MS files were matched within a window of +/− 1 Da to the precursor masses from the proteoform spectrum match (PrSM) table of the ProSightPD search results. Accession number was annotated for the proteoforms with matching masses and retention times within a window of +/− 1 min. The deconvoluted masses from both high- and low-resolution LC-MS were also matched within a window of +/− 1 Da against the library of theoretical masses of human β-casein proteoforms. Where possible, proteoforms were annotated based on both LC-MS/MS database search identification and on theoretical mass match. In cases where no identification was made by database search, proteoforms were annotated by matching of their experimental masses to theoretical masses with the condition that they eluted in a window of +/− 1 min from proteoforms identified by database search. Proteoform intensity was normalized on the sum of proteoform intensities for each protein.

## 5. Conclusions

Having a complete understanding of the human milk phospho- and glyco- proteome and endogenous peptidome is critical for a better understanding of the structure / function relationship of modified proteins and peptides. We have shown here that the endogenous peptidome is an unexplored source of PTM-rich peptides, wherein we have highlighted diverse and abundant phosphorylation and O-glycosylation of β-casein. Our work here lays a foundation for uncovering new PTMs in the human milk proteome and the endogenous peptidome that can be used in future studies to investigate functionality. 

## Figures and Tables

**Figure 1 ijms-22-08140-f001:**
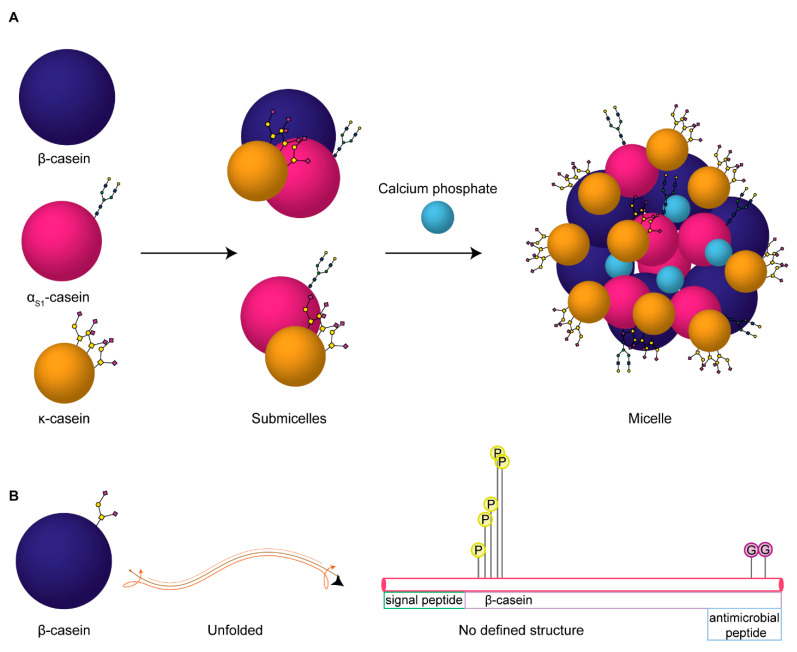
Proposed build-up of human milk casein micelles and proteoforms of casein proteins. (**A**) The three casein proteins in human milk, β-casein (dark purple), αS1-casein (magenta), and κ-casein (gold) are depicted according to their size and with known glycan structures annotated (as *N*- or *O*-glycans). These together interact to form submicellar particles, which constitute the building blocks of the casein micelles held together by colloidal calcium phosphate nanoclusters (light blue). (**B**) Our novel observation of β-casein as an *O*-glycoprotein, depicted as a spherical and unfolded structure. Lollipop depictions of the PTMs identified are shown on the relative sequence of the protein, where P indicates a phosphorylated residue and G indicates the newly discovered *O*-glycosylated residues; the height of the lollipop depictions is proportional to the relative abundances of the modifications identified on these sites.

**Figure 2 ijms-22-08140-f002:**
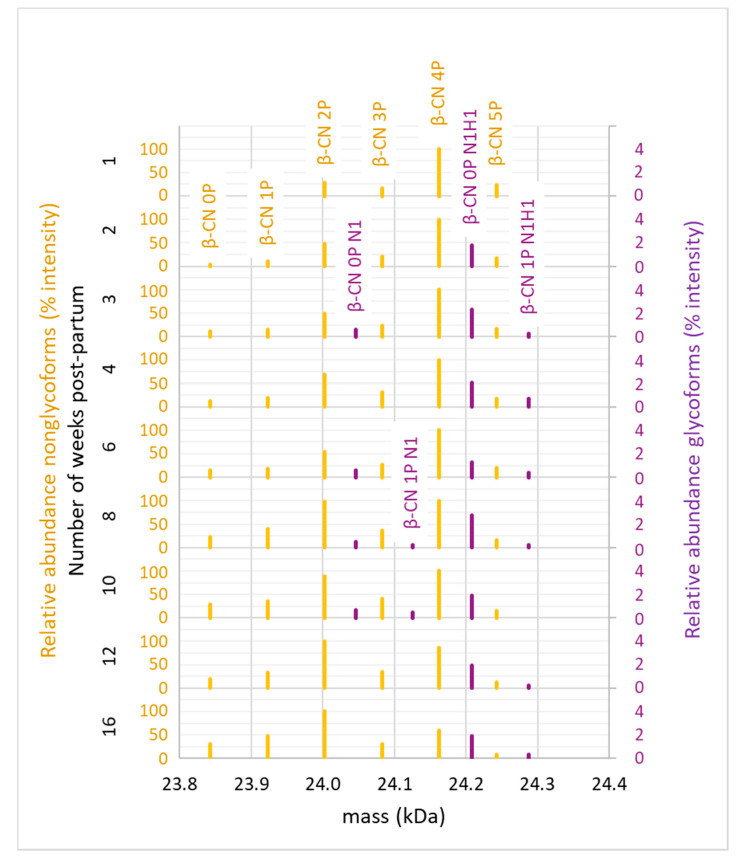
Deconvoluted mass spectra depicting the β-casein proteoforms identified in the whole milk of donor two throughout lactation. The non-glycosylated proteoforms (yellow) comprised phosphorylation in the range of 0–5P and were confirmed by database search of the intact protein LC-MS/MS results. Minor glycosylated proteoforms (~0–4% relative abundance of the most abundant β-casein proteoform; purple) could be annotated in the intact protein LC-MS data by the mass shift induced by the glycan residue masses.

**Figure 3 ijms-22-08140-f003:**
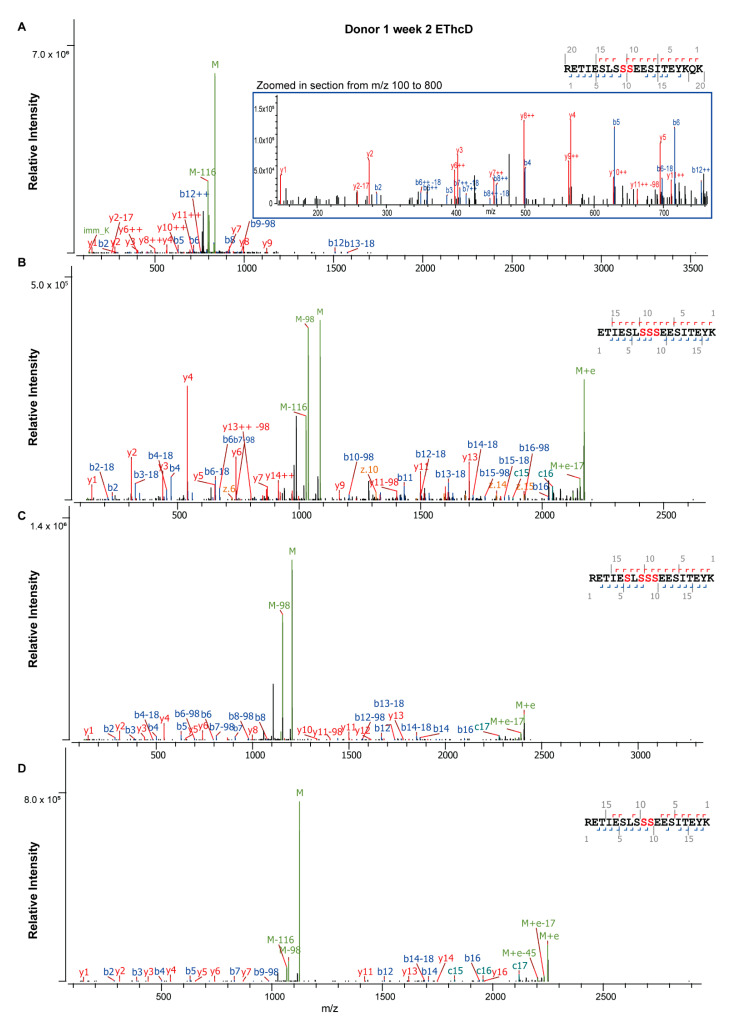
Illustrative EThcD fragmentation spectra of β-casein phosphopeptides. The N-terminal peptide _16_RETIESLSSSEESIT_30_ with up to seven phosphosites is depicted with varying degrees of site occupancy in the peptidome (**A**–**C**) and proteome (**D**). Designated phosphosites are indicated as red amino acids in the peptide sequence. Confidence in the phosphosite annotations are evident from the precursor mass with the neutral loss of phosphorylation from Ser (M−98 Da) upon fragmentation, with additional b- and y-ions from the peptide backbone placing the -98 Da at specific Ser residues. (**A**) Peptidome phosphopeptide harboring two phosphorylated Ser residues, the inset of *m/z* 100–800 shows the b- and y-ion series that might otherwise be obscured by the intensity of the other ions. (**B**) Peptidome peptide with three phosphorylated Ser residues. (**C**) Peptidome peptide with four phosphorylated Ser residues. (**D**) Proteome peptide with two phosphorylated Ser residues.

**Figure 4 ijms-22-08140-f004:**
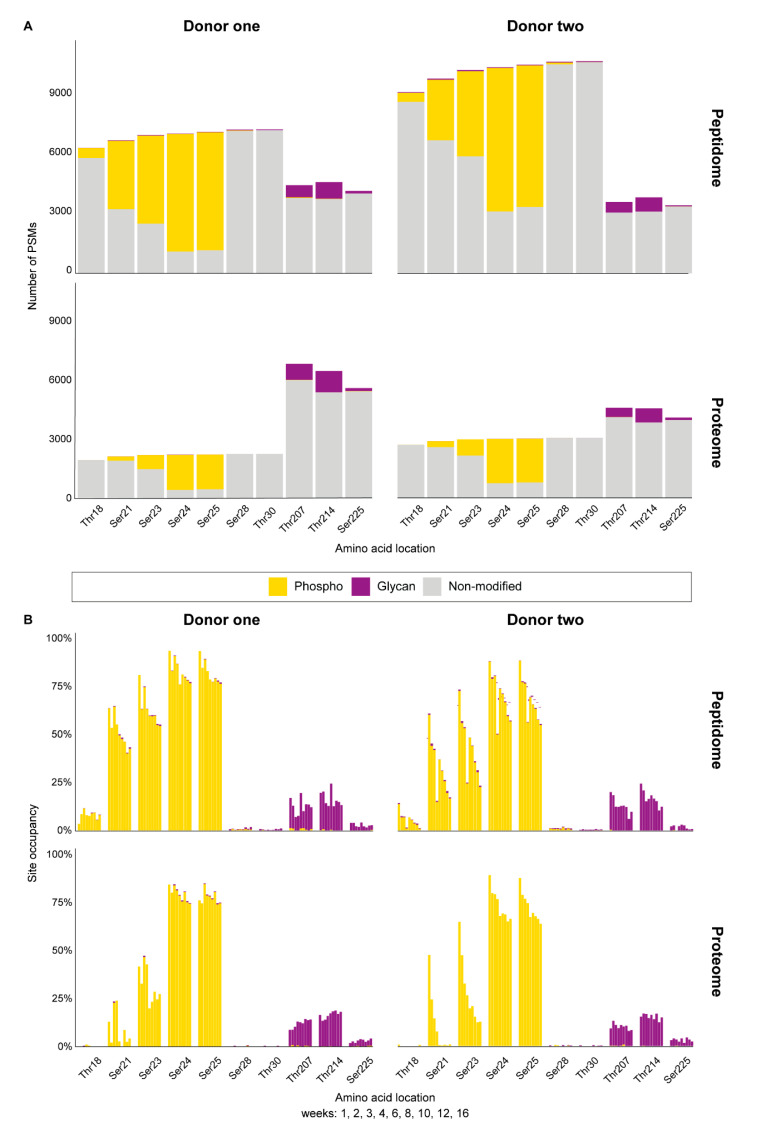
Semi-quantitative analysis of phosphorylated and glycosylated amino acids detected in β-casein across lactation. (**A**) Total number of detections of Thr18, Ser21, Ser23, Ser24, Ser25, Ser28, Thr30, Thr208, Thr214, and Ser225 for each donor in either the peptidome and proteome, colored for the presence of phosphorylation (yellow), glycosylation (purple), or without modifications (grey). (**B**) Percentage of modified amino acids amongst the total number of detections, separately displayed for lactation weeks 1, 2, 3, 4, 6, 8, 10, 12, and 16. While the observed PTMs are donor-specific, overall, the peptidome data displays a greater relative abundance of phosphorylated and glycosylated sites than the proteome data, and highly similar changes can be seen across lactation for both donors. Whereas all phosphorylation sites are near the N-terminus, the *O*-glycosylation sites are all at the C-terminus of β-casein.

**Figure 5 ijms-22-08140-f005:**
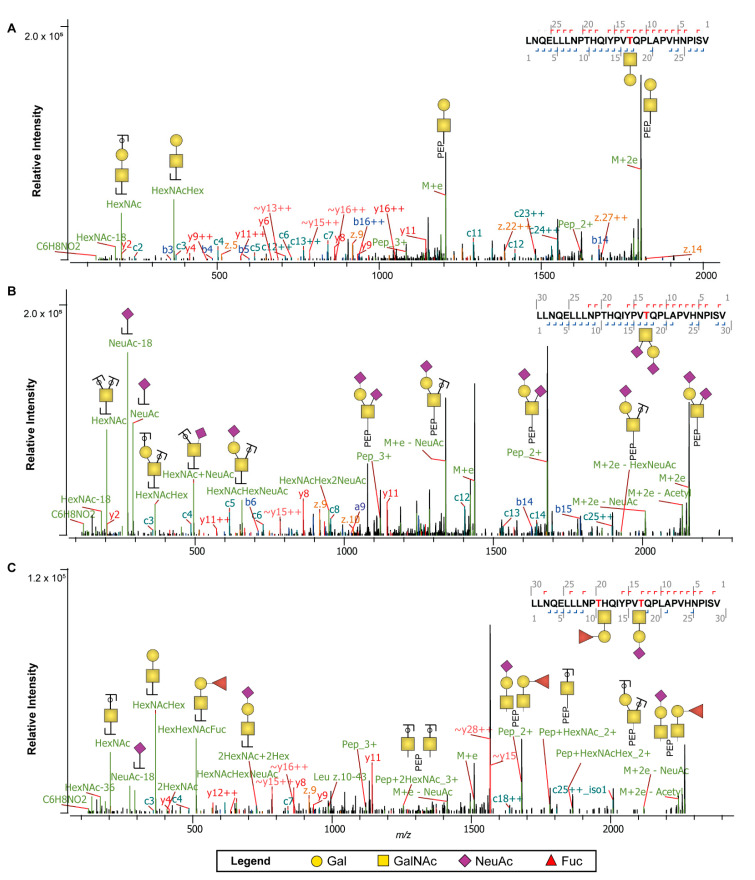
EThcD fragmentation spectra of the C-terminal β-casein peptide _197_LLNQELLLNPTHQYPVTQPLAPVHNPISV_226_ decorated by distinct *O*-glycans. (**A**) EThcD spectra with the *O*-glycopeptide carrying N_1_H_1_ glycosylation. (**B**) EThcD spectra with the *O*-glycopeptide carrying a sialylated glycan, N_1_H_1_S_2_. (**C**) EThcD spectra with the *O*-glycopeptide carrying sialylated and fucosylated glycans, N_1_H_1_S_1_ and N_1_H_1_F_1_. Note that a mixture of positional isomers could have been fragmented in the presented spectra and that different glycan structures likely exist for the displayed annotations. Only major glycan fragments are annotated. The monosaccharide legend is displayed at the bottom.

**Figure 6 ijms-22-08140-f006:**
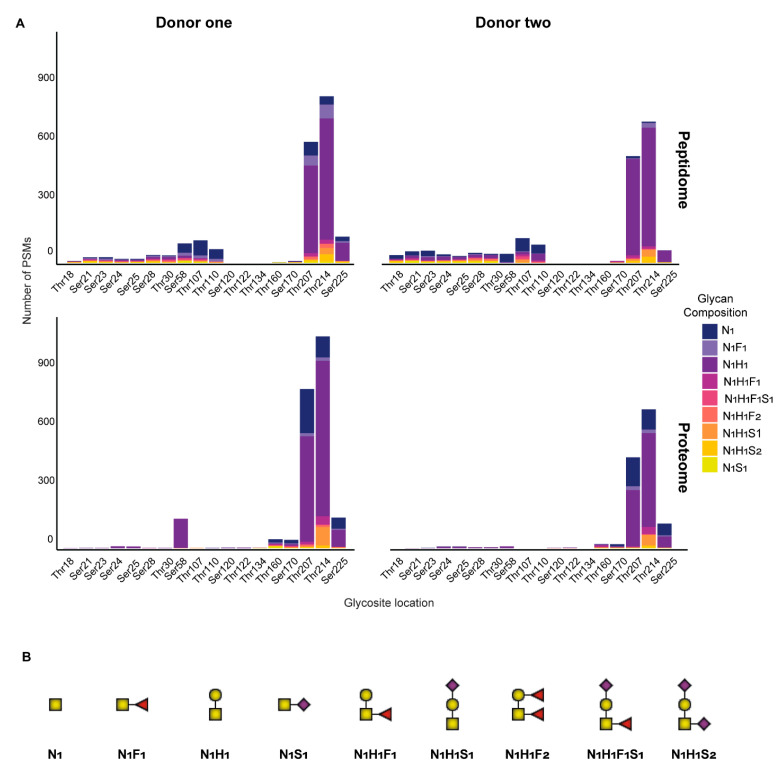
Site analysis of glycan species identified across the peptidome and proteome in each of the donors. (**A**) The glycan species are displayed for the peptidome and proteome at all possible Ser and Thr sites across the protein backbone for donors one and two. Glycan species are displayed as stacked bars of PSMs per site, with each glycan species represented with a different color. The sites Thr207 and Thr214 in the peptidome and proteome have the highest number of *O*-glyco-PSMs and are occupied by primarily N_1_ and N_1_H_1_ glycans. (**B**) Schematic representations of the nine different *O*-glycans identified on human β-casein. Our MS method did not distinguish structural elements of the glycosylation, and the representations only inform on the composition of the glycan species.

**Figure 7 ijms-22-08140-f007:**
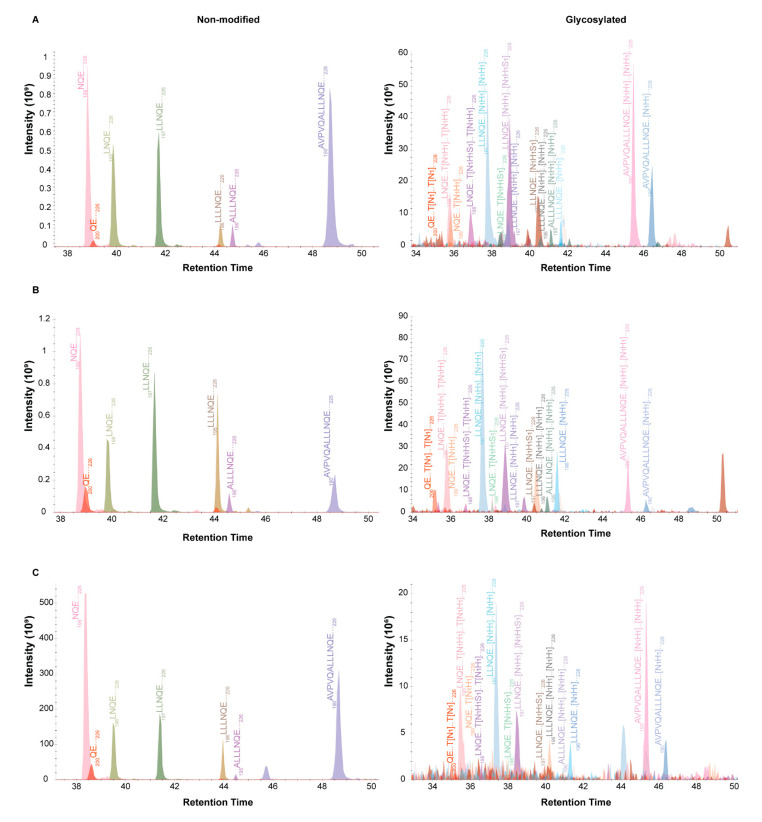
Changes in abundances of *O*-glycosylated peptide variants during early and late lactation. All presented ion traces originate from donor one and show the C-terminal β-casein ladder peptide series beginning with _190_AVPVQALLLNQ ELLLNPTHQIYPVTQPLAPVHNPISV_226_ and ending with the antimicrobial peptide _200_QELLLNPTHQYPVTQPLAPVHNPISV_226_. All ladder peptides are decorated by different *O*-glycan species, as containing a single glycan on Thr207 or Thr214, or having both sites occupied. All ladder peptides were found decorated with N_1_, N_1_H_1_ or N_1_H_1_S_1_ glycan species. Precursor ion traces are depicted as non-modified (left) and glycosylated (right). (**A**) MS1 traces of the differing glycopeptides relative to the non-modified peptide in week 1. (**B**) MS1 traces of the differing glycopeptides relative to the non-modified peptide in week 6. (**C**) MS1 traces of the differing glycopeptides relative to the non-modified peptide in week 16.

## Data Availability

The mass spectrometry proteomics data have been deposited to the MassIVE repository and can be accessed from the following link: ftp://massive.ucsd.edu/MSV000087464/. Additionally, all plotted LFQ non-glyco proteome and peptidome data has been made available for user interaction at the following link: (https://milkprofiling.hecklab.com/). All R-scripts have been published on https://github.com/hecklab/glyco-peptidome.

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
