# Peer review of "Monitoring Human Milk β-Casein Phosphorylation and O-Glycosylation Over Lactation Reveals Distinct Differences between the Proteome and Endogenous Peptidome"

_ijms, 2021, doi:10.3390/ijms22158140_

Round 1

Reviewer 2 Report

Nice piece of work that will certainly be publishable

I have a few comments that need to be addressed:

-article title should make it clear that this study is on human milk

-Figure 1 and description around it: this is very much a bovine milk view. How sure are you that this is also the care for human milk? Considering e.g., salt balance studies by Holt on human milk in the 1980’s, there appears to be very little inorganic phosphate associated with the casein fraction. Based on this, one could wonder whether the calcium phosphate clusters presented in Figure 1 are actually present in human milk. To my knowledge, no SAXS/SANS studies have been published confirming their presence, nor do TEM imagines of human milk casein micelles show electron dense granules like observed in bovine casein micelles. Refs. 16 and 17 quoted here also seem to be based on bovine and not human. Also good to keep in mind here that only the 3P-5P variants of human b-casein can actually interact with calcium phosphate particles, not the lower phosphorylation levels.

-Page 3: in terms of functions of the caseins, I would think supply of essential amino acids is the most important so why is this not mentioned

-Page 3: for the C-terminal peptide from b-casein: does this survive digestion in the GI tract? Not clear from the description

-Page 4: when you refer to 70% of human milk proteins in human milk are glycosylated, is this by mass %, by mol % or of the number of types of proteins found in human milk? Not clear

-Page 5: is 20% TCA the final concentration or the solution that was added? Needs to be clarified

-Page 8, top line: with new modifications I assume you are referring to the glycosylation, or the phosphorylation is in line with previous work. Good to clarify/specify

-Try to link changes in phosphorylation over time in Fig 2 to changes in salt content and partitioning, e.g., by Neville (Neville, M. C., Keller, R. P., Casey, C., & Allen, J. C. (1994). Calcium partitioning in human and bovine milk. Journal of dairy science77(7), 1964-1975.)

-Figure 4: hard to read. Is there an easier way to present this?

Round 2

Reviewer 1 Report

Quantification:
1. I am fully aware of the challenges that the authors have encountered for PTM occupancy determination and am not against using spectral counts to “represent” the abundances of different modification forms. The key point is to be transparent in the manuscript about the limitations of the method.
There is a correlation between peptide abundances and the resulting PSMs in an LC-MS/MS analysis. But to which degree this correlation allows the determination (or estimation) of PTM site occupancy needs more careful examination (i.e., the transition from Figure 4a to 4b). I agree with the authors that spectral counting works better when “really high numbers of spectral counts are detected”. However, is the approach still reliable when there were only tens of PSMs (Supplemental Table 5) to represent the site-specific glycosylation occupancy? Unfortunately, the presented data do not provide solid evidence to support that spectral-count-based site occupancy determination is accurate and reliable (see comment 2).
Besides, as pointed out by the authors, a portion of the PSMs still do not contain convincing fragments for a reliable site localization after stringent filtering (see comment 3). How these ambiguous PSMs affected the occupancy determination also needs further elaboration.
2. The statement “we compared MS1 integration with spectral counting for several (phospho/glyco)peptides, and generally arrived at a similar degree of PTM occupancy (approximately 5% of peptides, page 26).” needs further elaboration.
The manuscript presents the following data:
a. Figure 1 shows the relative abundances of various proteoforms - ~glycosylated forms showed relative abundances of ~1-3% ;
b. Supplemental Table 3 shows that ~5% of all the filtered PSMs could be attributed to glycosylation;
c. Supplemental Table 6 and figure 6 show PSM percentage of glycosylated T207/T214 to be 7-25%.
d. Supplemental Table 7 shows MS1 glycopeptide percentages of the ladder peptides to be 0-37%.
It is confusing that:
i) by the 5%, what exactly were the authors comparing? Overall protein glycosylation level or site-specific glycosylation occupancy?
ii) how exactly the authors access the so-called good agreement between MS1 integration and spectral counting?
iii) did the authors perform a similar comparison also for phosphopeptides?
Identification:
3. “The new O-glycosites we propose are limited to Thr207 and Thr214, which are already represented in Figure 5. The other Ser and Thr residues that could putatively be glycosylated do not display convincing fragmentation patterns and detections are very low in number of PSMs, and are not considered by us to be potential sites. Correspondingly, we have removed these disqualified detections from figure 4, so that our message is unambiguous.” I am surprised to see the statement that only glycosites T207 and T214 were confidently identified. Supplemental Table 5 indicates that there are hundreds of PSMs for glycosylated S58, T107, and S225 after stringent filtering. Does the statement mean that none of those PSMs could lead to confident identification (or glycan localization)? If this is the case, I do not understand why the authors still present those unreliable glycosites in the manuscript (Figures 4 and 6, and Supplemental Tables).
4. Figure 3A looks more convincing now. The same should apply to Figure 3B-D

Author Response

Response to reviewer’s comments

Reviewer 1

Quantification:

  1. I am fully aware of the challenges that the authors have encountered for PTM occupancy determination and am not against using spectral counts to “represent” the abundances of different modification forms. The key point is to be transparent in the manuscript about the limitations of the method.

There is a correlation between peptide abundances and the resulting PSMs in an LC-MS/MS analysis. But to which degree this correlation allows the determination (or estimation) of PTM site occupancy needs more careful examination (i.e., the transition from Figure 4a to 4b). I agree with the authors that spectral counting works better when “really high numbers of spectral counts are detected”. However, is the approach still reliable when there were only tens of PSMs (Supplemental Table 5) to represent the site-specific glycosylation occupancy?

Unfortunately, the presented data do not provide solid evidence to support that spectral-count-based site occupancy determination is accurate and reliable (see comment 2).

Answer: We are appreciative of the reviewer’s time and effort on reviewing our manuscript. We agree that the accuracy and reliability of PSM counting are important considerations going forward, as is the determination of the limit of quantification. While our current study design is not geared towards answering these questions specifically - we envision a separate work for this - we do try to be transparent about the quality and limitations of the work presented here.

For several glycopeptides we had already compared MS1 integration with PSM counting, but we understand that this comparison was obscured by spreading the information across tables and figures. To remedy this, we have added an additional table to the supporting information in which we directly compare the glycan occupancy values of the same peptides for the different methods. While the numbers are not identical they are certainly in the same range, and the lactational trends are conserved.

Besides, as pointed out by the authors, a portion of the PSMs still do not contain convincing fragments for a reliable site localization after stringent filtering (see comment 3). How these ambiguous PSMs affected the occupancy determination also needs further elaboration.

Answer: Since automatic annotation may lead to some misassignment, even with good quality scores, we consider it an important step to verify PTM sites with manual inspection of the fragment spectra. A site can be discarded entirely when no convincing spectra are present. For those sites that can be manually confirmed we do find that PSMs are consistently annotated by Byonic, along with the expected variations in peptide length and PTM composition. So, while we use manual verification to exclude sites, we do not use it to exclude individual spectra.

  1. The statement “we compared MS1 integration with spectral counting for several (phospho/glyco)peptides, and generally arrived at a similar degree of PTM occupancy (approximately 5% of peptides, page 26).” needs further elaboration.

Answer: We want to make our data analysis strategy as insightful as possible and have now included Supplemental Table 8 in which we directly compare the numbers and percentages for a consistent set of peptides. We hope this makes the quality of the data more clear to the reader and that it provides a framework to also interpret the other findings.

The manuscript presents the following data:

a. Figure 1 shows the relative abundances of various proteoforms - ~glycosylated forms showed relative abundances of ~1-3% ;

Answer: This is proteoform data, so only representative of the intact-protein data. For both donors and across time points, we detect lower PTM occupancies within the proteome than within the endogenous peptidome.

b. Supplemental Table 3 shows that ~5% of all the filtered PSMs could be attributed to glycosylation;

Answer: Indeed, from all identified PSMs from bottom-up proteomics an approximate 5% could be attributed to glycosylation. This percentage was derived for the proteome and peptidome data separately, incorporating both donors across all time points. So it is an average percentage for all samples.

c. Supplemental Table 6 and figure 6 show PSM percentage of glycosylated T207/T214 to be 7-25%.

Answer: Supplemental Table 6 and Figure 6 show PSM percentages of glycosylation on site Thr207 and Thr214. For donor one the peptidome data ranges from 7-16% for Thr207 and 13-25% for Thr214, and the proteome ranges from 14-19% and 8-14%, respectively. For donor two the peptidome data ranges from 6-20% for Thr207 and 11-25% for Thr214, and the proteome ranges from 8-13% and 13-17%, respectively. These values are visually depicted as stacked bar-charts, where the identified glycan compositions are displayed by number of identified PSMs, as total PSMs per donor and for the peptidome and proteome data across all possible Ser and Thr in the protein backbone. 

d. Supplemental Table 7 shows MS1 glycopeptide percentages of the ladder peptides to be 0-37%.

Answer: Supplemental Table 7 reports on the MS1 glycopeptide percentages for ladder peptides derived from the proteome and peptidome data, as percentages during specific weeks of lactation. For each ladder peptide there is also the possibility of differing glycan compositions occupying sites Thr207 and Thr214, with their respective percentages displayed.

It is confusing that:

i) by the 5%, what exactly were the authors comparing? Overall protein glycosylation level or site-specific glycosylation occupancy?

Answer: The stated 5% is an average glycosite occupancy from all samples across all lactational time points, and then derived for the proteome and the peptidome separately (b above). In addition to adding Supplemental Table 8 for the comparison, we have now made our phrasing clearer in lines 304 to 305 of the manuscript.

ii) how exactly the authors access the so-called good agreement between MS1 integration and spectral counting?

Answer: We have now included Supplemental Table 8 for the direct comparison between PSM counting and MS1 integration for the same peptides. We hope this makes it more apparent how the methods relate and how the results from PSM counting can be interpreted.

iii) did the authors perform a similar comparison also for phosphopeptides?

Answer: For phosphorylation we could relate our findings to the top-down experiments, as well as to the occupancy ratios that were already reported in the literature (unlike glycosylation). As such, we did not specifically compare spectral counting and MS1 area integration for phosphopeptides.

Identification:

  1. “The new O-glycosites we propose are limited to Thr207 and Thr214, which are already represented in Figure 5. The other Ser and Thr residues that could putatively be glycosylated do not display convincing fragmentation patterns and detections are very low in number of PSMs, and are not considered by us to be potential sites. Correspondingly, we have removed these disqualified detections from figure 4, so that our message is unambiguous.”

I am surprised to see the statement that only glycosites T207 and T214 were confidently identified. Supplemental Table 5 indicates that there are hundreds of PSMs for glycosylated S58, T107, and S225 after stringent filtering.

Does the statement mean that none of those PSMs could lead to confident identification (or glycan localization)? If this is the case, I do not understand why the authors still present those unreliable glycosites in the manuscript (Figures 4 and 6, and Supplemental Tables).

Answer: After manual inspection we indeed determined that there was no convincing fragmentation evidence to support glycosylation at  S58, T107, prompting the removal of the PSMs from Figure 4. In addition, we have now removed the PSMs from Supplemental Tables 5 and 6 as well. Even though fragmentation did not look convincing from site S225, due to the close proximity to Thr207 and Thr214 we could not justify excluding this site as well.

To further substantiate our decision process on removing the S58 and T107 we have added Supplemental Figure 2, displaying an clear example of a wrongfully annotated PSM. We have amended our text to also better explain data that were excluded, please see lines 319-328.

  1. Figure 3A looks more convincing now. The same should apply to Figure 3B-D

Answer: The spectra in Figure 3B-D actually contain many informative ions at lower as well as higher m/z ranges, actually warranting two insets. As we found this difficult to display in Figure 3 we have now included an additional Supplemental Figure 3 in which we show the separate low and high insets.